

# Metagenomic survey of methanesulfonic acid (MSA) catabolic genes in an Atlantic Ocean surface water sample and in a partial enrichment

Ana C. Henriques,  Rui M.S. Azevedo and  Paolo De Marco

Instituto de Investigação e Formação Avançada em Ciências e Tecnologias da Saúde (IINFACTS), CESPU, Gandra PRD, Portugal

## ABSTRACT

Methanesulfonic acid (MSA) is a relevant intermediate of the biogeochemical cycle of sulfur and environmental microorganisms assume an important role in the mineralization of this compound. Several methylotrophic bacterial strains able to grow on MSA have been isolated from soil or marine water and two conserved operons, *msmABCD* coding for MSA monooxygenase and *msmEFGH* coding for a transport system, have been repeatedly encountered in most of these strains. Homologous sequences have also been amplified directly from the environment or observed in marine metagenomic data, but these showed a base composition (G + C content) very different from their counterparts from cultivated bacteria. The aim of this study was to understand which microorganisms within the coastal surface oceanic microflora responded to MSA as a nutrient and how the community evolved in the early phases of an enrichment by means of metagenome and gene-targeted amplicon sequencing. From the phylogenetic point of view, the community shifted significantly with the disappearance of all signals related to the *Archaea*, the *Pelagibacteraceae* and phylum SAR406, and the increase in methylotroph-harboring taxa, accompanied by other groups so far not known to comprise methylotrophs such as the *Hyphomonadaceae*. At the functional level, the abundance of several genes related to sulfur metabolism and methylotrophy increased during the enrichment and the allelic distribution of gene *msmA* diagnostic for MSA monooxygenase altered considerably. Even more dramatic was the disappearance of MSA import-related gene *msmE*, which suggests that alternative transporters must be present in the enriched community and illustrate the inadequacy of *msmE* as an ecofunctional marker for MSA degradation at sea.

Subjects Ecology, Environmental Sciences, Marine Biology, Microbiology, Molecular Biology
Keywords Methanesulfonic acid, Sulfur, Biogeochemical cycle, Ocean, Bacteria, Gene, Metagenomics

# INTRODUCTION

It is known that methanesulfonic acid (MSA) has been produced during millennia in the atmosphere by the oxidation of dimethylsulfide (DMS) that escapes from the  seawater surface (*Andreae, 1986*; *Hynes, Wine & Semmes, 1986*; *Mihalopoulos et al., 1992*; *Koga & Tanaka, 1993*; *Kelly & Murrell, 1999*). DMS is mainly a byproduct of the degradation

Corresponding author
Paolo De Marco,
paolo.marco@iscsn.cespu.pt

of marine photosynthetic organisms (*Todd et al., 2011*) and is the major component of marine emissions of volatile sulfur (*Gondwe et al., 2003*). Due to the huge scale of this biogeochemical process, very significant amounts (est. $10^{10}$ kg) of MSA form annually and deposit back onto the sea or land surfaces (*Charlson et al., 1987*; *Kelly & Murrell, 1999*; *Gondwe et al., 2003*). MSA has not been found accumulating in any environment (apart from perennial ices (*Legrand & Feniet-Saigne, 1991*; *Whung et al., 1994*)), which means that this compound is readily degraded in nature. Several methylotrophic bacterial strains have been isolated that can use MSA as sole source of carbon and energy (*Kelly & Baker, 1990*; *Thompson, Owens & Murrell, 1995*; *De Marco et al., 2000*; *De Marco et al., 2004*; *Baxter et al., 2002*; *Moosvi et al., 2005b*) while other microbes are known to use this molecule just as a sulfur supply (*Kelly & Murrell, 1999*). An operon (*msmABCD*) encoding a heteromeric monooxygenase (MSA monooxygenase, or MSAMO) and another operon (*msmEFGH*) encoding uptake proteins have been found in several MSA-utilizing strains (*De Marco et al., 1999*; *Baxter et al., 2002*; *Jamshad et al., 2006*; *Henriques & De Marco, 2015a*; *Henriques & De Marco, 2015b*) and in a marine bacterium isolated from Western Pacific surface waters (*Oh et al., 2010*).

All proteobacterial strains known to use MSA methylotrophically that have been analyzed at the molecular level have shown to possess gene *msmA* and most of them carry gene *msmE* (*Henriques & De Marco, 2015a*; *Henriques & De Marco, 2015b*). By contrast, the only Actinobacterial strain known to grow on MSA as a carbon source (*Rhodococcus* str. RD6.2 (*De Marco et al., 2004*)) harbors an alternative gene, *ssuD*, coding for an enzyme previously associated solely to non-methylotrophic MSA utilization (*Eichhorn, Van Der Ploeg & Leisinger, 1999*; *Endoh et al., 2003*), plus some broad-range alkanesulfonate monooxygenases (*Henriques & De Marco, 2015c*). Homologs of the *msm* genes and operons have also been directly amplified from environmental DNA (*Baxter et al., 2002*; *Henriques & De Marco, 2015a*), retrieved by metagenomic sequencing projects (*Leitão, Moradas-Ferreira & De Marco, 2009*) or found highly expressed in metatranscriptomic analyses of surface seawater (*Gifford et al., 2013*). Dozens of hits for *msmA* gene can also be recovered from the recently published *Tara* Oceans project data (*Sunagawa et al., 2015*) and one apparent *msm* double operon was found in one of the single-cell genomes (*Alphaproteobacterium* SCGC AAA536-B06) sequenced from the Mediterranean Sea (*Swan et al., 2013*). Among the MSAMO enzyme components, the ferrodoxin (MsmC) and the FAD-binding NADH-dependent reductase (MsmD) are very similar to analogous components of unrelated oxygenases while MsmB, like many other examples of hydroxylase beta subunits, shows poor sequence conservation (*De Marco et al., 1999*; *Baxter et al., 2002*). On the contrary, alpha subunits of MSAMO hydroxylases (MsmA) show strong conservation and a peculiar 26-amino acid-long spacer within the Rieske-type [2Fe-2S]-binding motif. Among the polypeptides involved in the import of MSA into the cell, MsmE is the one that, in the proposed model, binds MSA in the periplasm. For these reasons, genes *msmA* and *msmE* have been selected as promising markers for MSA-utilizing bacteria in the environment. However, most likely owing to lower sequence conservation, designing robust primer pairs for gene E and obtaining bona fide *msmE* amplicons has been much less successful than with gene *A* (*Henriques & De Marco, 2015a*). In general, most of these genes obtained

directly from marine water samples were low or very low in G + C content (36–48%) while cultivated strains obtained from both soil, estuary and seawater had G + C content levels in the 47–66% range. This dichotomy is not unique to MSA-utilizers or *msm* genes: the single-cell genomes retrieved by *Swan et al. (2013)* showed clear signs of streamlining and much lower GC% than cultivated marine strains, which may be a reflection of the generalized oligotrophic traits of unculturable bacteria versus the copiotrophic nature of cultivated types. Consistent with all these observations is the idea that laboratory isolates are no proper representatives of natural populations and that the only way to obtain a truthful picture of microbial communities is through *in situ* physiological observation and/or direct molecular investigation.

In this work, we analyzed phylogenetically and functionally a surface coastal seawater sample from the Atlantic Ocean and repeated the analysis after partial enrichment with MSA as sole added source of carbon, energy and sulfur. Amplicon survey analyses on functional markers for MSA degradation, genes *msmA* and *msmE*, were also carried out.

## MATERIAL & METHODS

### Seawater sample collection and metagenomic DNA isolation

Atlantic Ocean surface water was collected on Dec 3rd, 2014 along the coast of Leça da Palmeira, Portugal (approximate coordinates 41.226956, −8.720528). Approximately 15 L of seawater were collected off the rocky shore at rising tide into clean bottles, which were immediately transported to the lab in an isothermal bag with ice packs. Before starting the procedures, the bottles were shaken in order to homogenize the samples. Three fractions of 0.5 L were filtered through 1.2 μm glass fiber filters, which were immediately wrapped in aluminum foil and stored at −80 °C for later quantification of chlorophyll. The determination of chlorophyll *a* concentration was performed through a spectrophotometric method as previously described (*Inag, 2009*). Chlorophyll *a* concentration was calculated through the Jeffrey and Humphrey equation (*Jeffrey & Humphrey, 1975*). Measures for pH and conductivity were also performed. Two fractions of 5 L of the seawater sample were filtered in parallel through 1.2 μm, 0.45 μm and 0.2 μm filters in succession. The filters from one 5 L fraction were immediately used for DNA extraction using the PowerWater DNA Isolation Kit (MO BIO Laboratories, Inc.) according to the manufacturer's instructions.

### Marine enrichment with MSA and metagenomic DNA isolation

The biomass retained on the filters from the second 5 L fraction was resuspended in the last 200 mL of the ocean water sample. The filters were removed and 1 mL of alkaline (pH 8) sodium methanesulfonate (MSA) 1 M was added to the suspension (5 mM final concentration). Similar amounts of alkaline MSA were used to spike the enrichment at days 7, 9 and 12. The suspension was incubated aerobically at room temperature (ca. 20 °C) in the dark, continuously mixed by a magnetic stirrer. This schedule was deliberately maintained during 16 days, with no subculturing, in order to produce just a partially enriched culture. The biomass at the end of the enrichment process was collected by centrifugation and DNA was extracted with PowerWater DNA Isolation Kit (MO BIO Laboratories, Inc., with adaptations to accommodate biomass in a pellet rather than on a filter). For convenience,

the seawater sample (time 0) and the corresponding enrichment (E) culture will be designated SCD0 and SCDE, respectively. This study was limited to the observation of the evolution of the microbial community of a single sample and as such the following results are to be considered exploratory.

## Whole metagenome sequencing

The metagenomes (from SCD0 and SCDE samples) were sequenced at Molecular Research LP (Shallowater, TX, USA). Paired-end sequencing libraries were prepared (2 × 101 bp) and sequencing was performed using the Illumina HiSeq2500 platform. The libraries were prepared using Nextera DNA Sample preparation kit (Illumina) following the manufacturer's user guide. The initial concentration of DNA was evaluated using the Qubit® dsDNA HS Assay Kit (Life Technologies). The samples were then diluted accordingly to achieve the recommended DNA input of 50 ng at a concentration of 2.5 ng/µL. Subsequently, the samples underwent fragmentation, addition of adapter sequences and PCR amplification (5 cycles) during which a unique index was added to each sample. The average library size was determined using the Agilent 2100 Bioanalyzer (Agilent Technologies). The libraries were then pooled in equimolar ratios at 2 nM and 5 µL of the library pool was clustered using the cBot (Illumina) and sequenced paired-end for 200 cycles using the HiSeq 2500 system (Illumina).

The quality of the sequencing reads from both libraries was assessed using FastQC on Galaxy web-based platform (https://usegalaxy.org/). The libraries were also checked for human contamination with Kraken Metagenomics (*Zaharia et al., 2011*; *Wood & Salzberg, 2014*) at Illumina BaseSpace (http://basespace.illumina.com/home/index). As the resulting values for the presence of human sequences were very low (0.15% for sample SCD0 and 0.04% for sample SCDE), no filtering was performed before the subsequent steps of analysis (general statistics are described in Table S1).

Sequencing data for the two samples, SCD0 and SCDE, were submitted to the European Nucleotide Archive (http://www.ebi.ac.uk/ena) under project number PRJEB9018 and sample accession numbers ERS700852 and ERS700853, respectively. The analyses of the metadata were performed through the EBI Metagenomics service pipeline (https://www.ebi.ac.uk/metagenomics/pipelines/2.0 (*Mitchell et al., 2015a*)) that includes a quality control step, a taxonomic analysis step based on 16S rDNA sequences and a functional analysis of predicted protein coding sequences using the InterPro resource (*Mitchell et al., 2015b*).

BIOM (see http://biom-format.org/) files containing phylogenetic classification information provided by EBI metagenomics were used to construct rarefaction curves using MEGAN (version 5.10.6; *Huson, Mitra & Ruscheweyh, 2011*). Phylogenetic data were also used to estimate the alpha diversity of the two samples (Shannon index (*Shannon, 1948*), evenness (*Mulder et al., 2004*) and Chao species estimator (*Chao, 1984*)). For beta diversity analysis, Jaccard (*Jaccard, 1912*), Kulczynski (*Faith, Minchin & Belbin, 1987*), and Chao (*Chao, Chazdon & Shen, 2005*) indices were calculated through the Vegan package (*Oksanen et al., 2015*). Bray-Curtis dissimilarity index (*Bray & Curtis, 1957*) was calculated starting from relative abundances. Only significant differences in phylogenetic or functional composition were retained (Fisher's exact / $\chi^2$ test with a significance level of 0.05 with a

Bonferroni correction on the number of comparisons, in order to minimize false positives; code was adapted from Metastats (*White, Nagarajan & Pop, 2009*)).

Assembled metagenomic data from both samples were also submitted to the DOE Joint Genome Institute's Integrated Microbial Genome Metagenomic Expert Review (IMG/MER) annotation pipeline (http://img.jgi.doe.gov/) for functional and taxonomic annotation (*Markowitz et al., 2014*) (SCD0: GOLD Analysis Project Id Ga0069134/biosample ID Gb0111627; SCDE: GOLD Analysis Project Id Ga0069135/biosample ID Gb0111630). Before submission, reads were quality checked (FastQC for quality control at Galaxy for trimming and filtering): only sequences with quality scores equal or higher than 20 over 95% or more of the nucleotides were kept. For each sample, forward and reverse sequence files were merged and assembled using Megahit (*Li et al., 2015*) (general statistics are reported in Table S2). A further analysis was performed at MG-RAST (*Meyer et al., 2008*) using the subsystems approach (*Overbeek et al., 2005*): metagenome SCD0 was submitted under sample no 4698364.3 and SCDE under sample no 4698363.3. Binning was performed using MetaBat v0.25.4 (*Kang et al., 2015*) and the bins obtained were analyzed by MG-RAST.

A flowchart with the major steps of the analysis is available in Fig. S1.

## msmA and msmE amplicon surveys

In order to obtain DNA amounts sufficient for the subsequent analyses, the REPLI-g® MiniKit (QIAGEN) was used to amplify the DNA from each metagenomic sample, according to the manufacturer's instructions. Negative controls starting from ultrapure nuclease-free water yielded no product.

The amplified metagenomes were tested with primers directed to genes *msmA* and *msmE*. As amplicons with less than 400 bp were needed for Ion Torrent sequencing, primer pairs SarA139fwd/SarA488rev and SarE322fwd/SarE704rev (Table S3) were employed in order to obtain amplicons with 349 bp and 382 bp, respectively. However, when amplification was performed directly with these primers, the signals obtained were too weak or absent. As such, a nested PCR approach was carried out. The first round was performed using primer sets SarA124fwd/SarA1053rev or SarE133fwd/SarE1125rev for *msmA* and *msmE* amplicons, respectively (Table S3). Some parameters had to be adjusted in order optimize the reactions, and the successful conditions are reported in Table S4. Negative controls received PCR water instead of DNA. Positive controls contained DNA from Sargasso Sea Metagenome clone EF103447 (*Leitão, Moradas-Ferreira & De Marco, 2009*). Only primers directed to the low-GC Sargasso Sea Metagenome *msm* sequences had previously been successful at amplifying these genes from a seawater metagenomic sample from the same location (*Henriques & De Marco, 2015a*). Accordingly, the primers used in this work were based on the known Sargasso Sea Metagenome *msm* sequences. Amplification products were submitted for sequencing at Stabvida Lda. (Caparica, Portugal). After determining the exact PCR product concentrations with Qubit 2.0 Fluorometer (Invitrogen) and Qubit dsDNA BR Kit (Invitrogen), platform-specific barcoded adapters were added to each sample in the preparation of libraries with KAPA Library Preparation Kit (Kapa Biosystems) and NEXTflex™ DNA Barcodes for Ion PGM (Bioo Scientific). Samples

were sequenced on an Ion Torrent Personal Genome Machine (PGM, Life Technologies, Thermo Fisher Scientific) using Ion PGM$^{TM}$ Hi-Q Sequencing Kit reagents. Reads were provided free from barcodes and adapters sequences.

Sequencing data from *msmA* amplicons, SCD0-A and SCDE-A sets, and *msmE* amplicons, SCD0-E set, were submitted to the European Nucleotide Archive (http://www.ebi.ac.uk/ena) under sample accession numbers ERS954926, ERS954925 and ERS954927, respectively, within the metagenomic study PRJEB9018.

The overall quality scores of the output data from the Ion Torrent sequencing was assessed using FastQC on Galaxy web-based platform. Sequencing reads were then trimmed and filtered based on length and quality. For the *msmA* amplicon, reads ranging from 300 to 365 bp were selected; for the *msmE* amplicon, sequences with length between 335 to 420 bp were retained. In the quality filtering step, sequences with quality lower than 20 over more than 15% of the nucleotides were discarded. Chimeric sequences were removed with Chimera Check (*Edgar et al., 2011*) available in the FunGenePipeline (*Fish et al., 2013*). The resulting files were submitted to FrameBot (*Wang et al., 2013*) for translation and frameshift correction (FunGenePipeline). Sequences containing stop and/or undefined codons were discarded (general statistics are reported in Table S5). For each dataset, protein sequences were aligned with hmmalign (HMMER3 (*Eddy, 2011*)) and clustered by complete linkage clustering (mcClust (*Loewenstein et al., 2008*), FunGenePipeline). Rarefaction (FunGenePipeline (*Fish et al., 2013*)) was based on data from the clust file at 0.03 distance. Conservation analysis of the MsmA and MsmE predicted protein sequences was performed on alignments generated by hmmalign (HMMER3) and analyzed by Jalview (*Waterhouse et al., 2009*). Alpha diversity analysis (Shannon index, evenness and Chao species estimator) was performed on clustering values obtained at 0.03 distance for each set of results. Beta diversity (Bray-Curtis, Jaccard, Kulczynski, and Chao indices) between SCD0-A and SCDE-A sequence sets was computed on the clustered (0.03 distance) aligned MsmA predicted sequences. For Bray-Curtis, relative cluster abundances were employed. The representative sequence for each cluster was obtained using Representative Sequences (FunGenePipeline). In this case, distance cutoff values were chosen in order to obtain in the region of 20–30 clusters from each dataset (0.15 for SCD0-A/SCDE-A and 0.10 for SCD0-E). The nucleotide sequences corresponding to these cluster-representative sequences were retrieved and used to infer phylogenetic trees by Maximum Likelihood with 100 bootstrap iterations (as implemented in PhyML (*Guindon et al., 2010*) at Phylogeny.fr (*Dereeper et al., 2008*)). A flowchart with the major steps of this analysis is available in Fig. S2.

## RESULTS

### Characterization of the samples

The seawater sample collected for this study was at a temperature of 15.5 °C and 3.34% salinity (conductivity 47.9 mS/cm). The pH value was 8.07 and the concentration of chlorophyll *a* was 1.06 mg/m$^3$.

Optical density (at 600 nm) of the filtered sample was 0.606 at time 0 (SCD0), dropped to 0.415 at day 2 and rose towards the last days of incubation to 0.758 (SCDE). During the enrichment process the suspension progressively lost its original greenish hue.

 

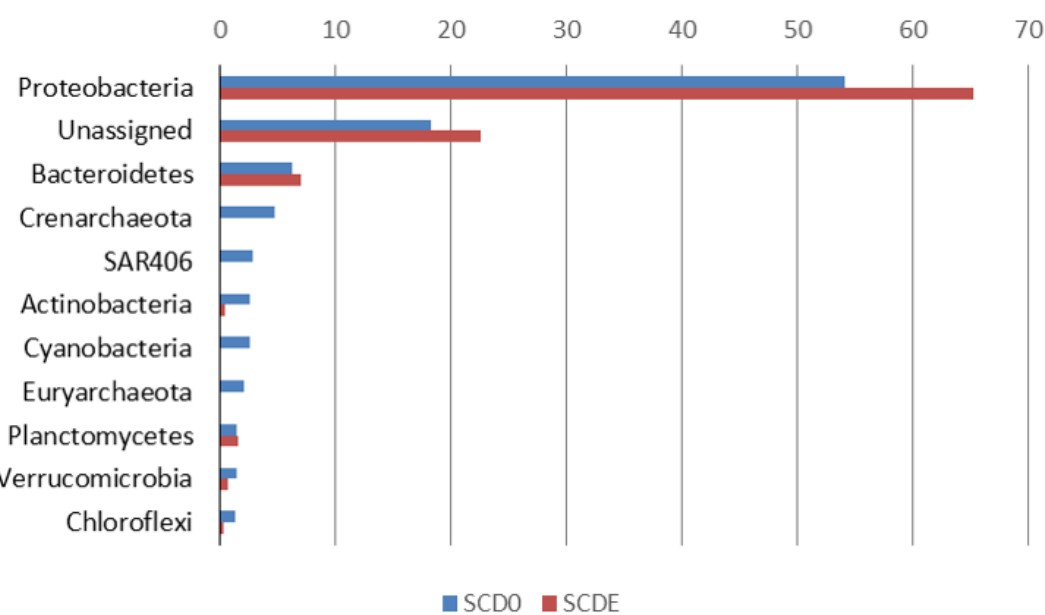

**Figure 1** **Phylogenetic composition (phyla) of the two metagenomes, SCD0 and SCDE (in percentage).**
Only phyla with abundance ≥1% in either sample are shown (based on EBI Metagenomics analysis of the data).

## Analysis of phylogenetic composition

Despite the fact that sample SCDE underwent just a partial enrichment with no dilution or subculturing, this was enough for the G + C content of the community's metagenome to shift from 43.30% (SCD0) to 53.29% (SCDE).

Phylogenetic composition analysis was performed by EBI Metagenomics on the metagenomic data, based on prokaryotic rRNA gene sequences. Rarefaction curves were generated from these results for both metagenomes (Fig. S3). These curves show that a reasonable coverage of the communities' composition was achieved: however, neither reaches a definite plateau meaning that deeper sequencing levels would have been desirable.

Phylogenetically, the community before the enrichment was composed primarily by taxa typical of surface seawater, namely *Alphaproteobacteria* of the *Pelagibacteraceae* and the *Rhodobacteraceae*, *Gammaproteobacteria* of the *Halomonadaceae* and *Piscirickettsiaceae*, *Archaea* of the *Crenarchaeota* (with prominence for *Nitrosopumilus*) and the *Euryarchaeota* (*Thermoplasmata* Marine group II), *Bacteroidetes* of the *Flavobacteriaceae* and the uncultured marine phylum SAR406 (Figs. 1 and 2).

Clearly, some of these groups declined or disappeared during our enrichment program (Figs. 1, 2 and 3): all the *Archaea* and SAR406, the *Pelagibacteraceae*, *Halomonadaceae*, *Flavobacteriaceae* (a family known to harbor methylotrophic species (*De Marco et al., 2004*; *Moosvi et al., 2005a*; *Boden et al., 2008*; *Madhaiyan et al., 2010*)) and *Variovorax paradoxus* (*Comamonadaceae*) a known methylotroph (*Anesti et al., 2005*). Other groups thrived: the *Rhodobacteraceae* and the *Hyphomonadaceae* within the *Alphaproteobacteria*, the *Oceanospirillaceae*, the *Piscirickettsiaceae*, two families of *Alteromonadales* and genus *Alcanivorax* within the *Gammaproteobacteria*, the *Saprospiraceae* within the *Bacteroidetes*. It

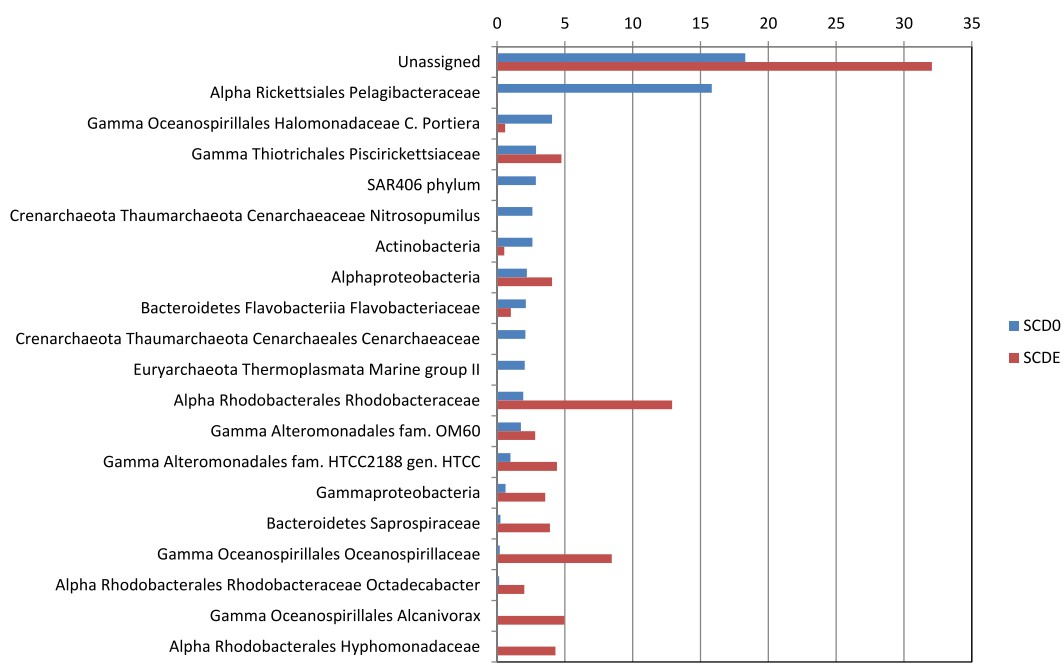

**Figure 2 Taxonomic composition of the two metagenomes (in percentage).** Only taxa with abundance ≥2% in either sample are shown. Taxonomic classification as provided by the phylogenetic analysis of the EBI Metagenomics pipeline (which explains some apparent taxonomic inconsistency such as "Alpha Rickettsiales Pelagibacteraceae," referring to sequences that could be classified down to the family level, and further down "Alphaproteobacteria," referring to sequences that could be classified just at the class level). Alpha, and Gamma are abbreviations for the corresponding classes within the *Proteobacteria*.

is worth noticing that: fam. *Rhodobacteraceae* includes several methylotrophic species and namely marine MSA-degrader *Marinosulfonomonas* (*Thompson, Owens & Murrell, 1995*); fam. *Piscirickettsiaceae* includes the genus *Methylophaga*, a known methylotrophic taxon relevant in surface sea waters (*Neufeld et al., 2008*). Indeed, the signal for *Methylophaga* itself at the genus level shot up from below detection to 82 hits; genus *Alcanivorax*, which contains marine representatives that have previously been found in a dimethylsulfide-based enrichment (*Schäfer, 2007*) and one known potentially methylotrophic species (*A. borkumensis* strain SK2—http://www.ebi.ac.uk/biomodels-main/BMID000000083743). Despite the fact that to our knowledge no methylotrophic species is known among the *Hyphomonadaceae*, the considerable increase in its abundance after the enrichment (225x) seem to suggest a possible involvement of members of this group in the turnover of MSA. Other rarer taxa that showed sizeable abundance increases after the enrichment were the *Cryomorphaceae* (*Bacteroidetes*), among the *Gammaproteobacteria* several branches within the *Alteromonadales*, several groups within the *Oceanospirillales* like the *Oceanospirillaceae* (and species *Neptuniibacter caesariensis*), the *Oleiphilaceae*, genus *Halomonas*, and rather surprisingly the *Enterobacteriaceae*. Notably, known methylotrophic-containing taxa like the *Hyphomicrobiaceae* (*Alphaproteobacteria*) and *Methylotenera mobilis* (*Betaproteobacteria*) also increased significantly.

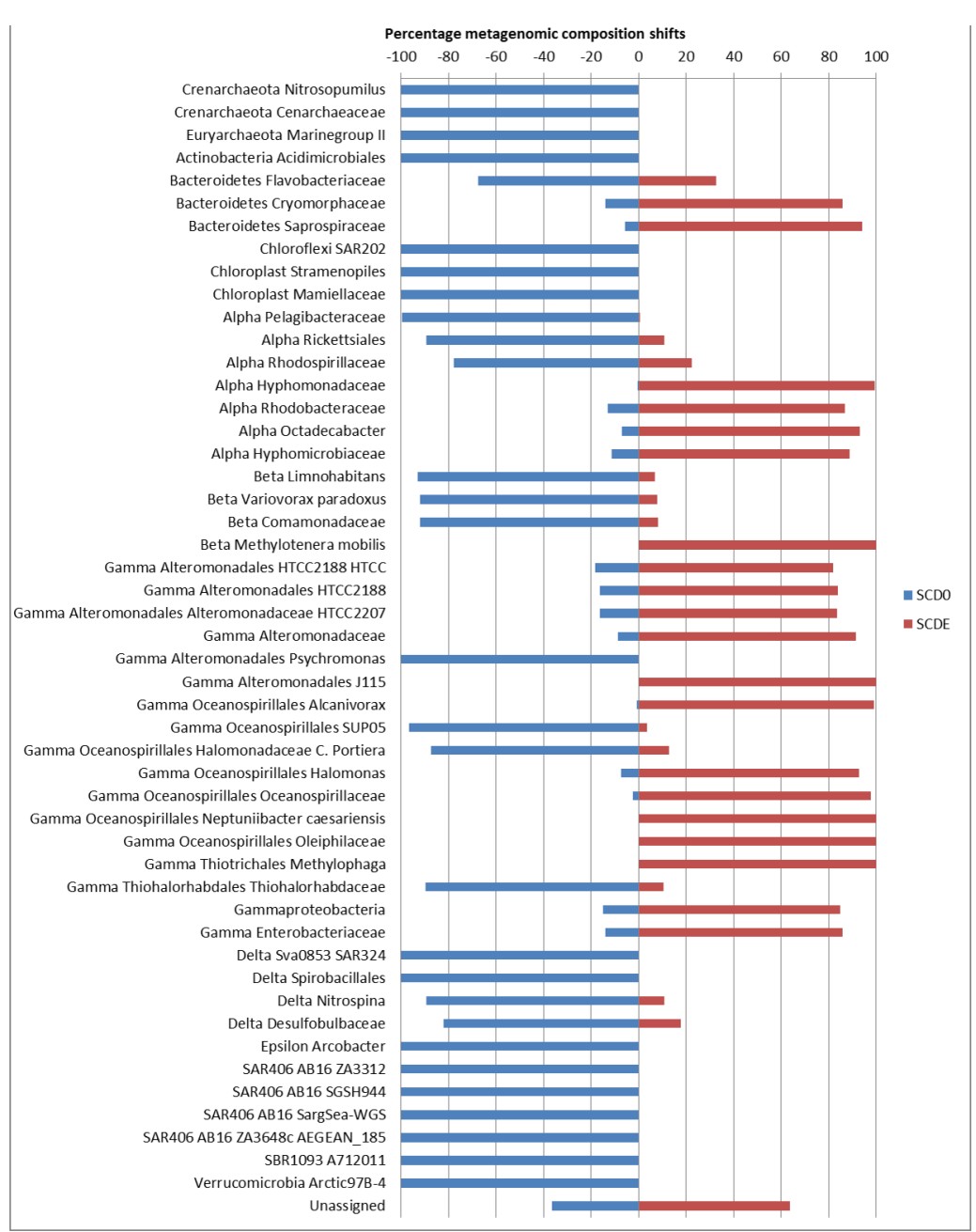

**Figure 3  Significant shifts in phylogenetic composition observed due to the enrichment.** Shown are percentages of the abundance of each taxon in each sample (SCD0 or SCDE) over the total taxon abundance (SCD0 + SCDE). Only statistically significant differences are shown. Taxa are as provided by the phylogenetic analysis of the EBI Metagenomics pipeline (which explains some apparent taxonomic inconsistency, see note in Fig. 2). Alpha, Beta, Gamma, Delta and Epsilon are abbreviations for the corresponding classes within the *Proteobacteria*.

Table 1 **Alpha diversity results for samples SCD0 and SCDE based on phylogenetic data.**

| Alpha diversity indicators | SCD0 | SCDE |
|---|---|---|
| Shannon index (H') | 3.95 | 4.97 |
| Evenness (E) | 0.68 | 0.84 |
| Number of observed taxa | 329 | 363 |
| Chao taxa number estimator | 460.8 | 544.4 |

Table 2 **Beta diversity between samples SCD0 and SCDE based on phylogenetic data.**

| Beta diversity indexes | |
|---|---|
| Bray-Curtis | 0.56 |
| Jaccard | 0.76 |
| Kulczynski | 0.60 |
| Chao index | 0.77 |

Metagenomic data binning by MetaBat failed to generate discrete bins with sample SCD0: this is not surprising with a diverse natural community under no definite selective pressure where no single species dominate. However, three bins were obtained with sample SCDE which by and large matched the results obtained by direct metagenomics analysis. Bin 1, assigned to an *Alcanivorax* sp. strain, contains genes coding for diagnostic enzymes for the assimilation of $C_1$ carbon through the serine cycle (serine hydroxymethyltransferase, serine-glyoxylate aminotransferase and isocitrate lyase). Bin 2, tentatively classified as the genome of a *Hyphomonas* sp. strain, also contains genes for the serine cycle (serine hydroxymethyltransferase) and for the anaplerotic ethylmalonyl-CoA and glyoxylate pathways (propionyl-CoA carboxylase, ethylmalonyl-CoA mutase and isocitrate lyase). Bin 3 showed phylogenetic markers mostly associated with genus *Methylophaga*: it contains genes diagnostic for the ribulose monophosphate (RuMP) cycle (3-hexulose-6-phosphate synthase and 6-phospho-3-hexuloisomerase), an alternative metabolic route for the assimilation of $C_1$ carbon, which is the typical metabolic route found in *Methylophaga* species.

Alpha and beta diversity estimates were based on the phylogenetic data obtained by the EBI Metagenomics analysis. The values are listed in Tables 1 and 2. The effects of the enrichment were an increase in the number of taxa observed, internal diversity (H'), and evenness. This was due to the sharp decline or disappearance of some of the major taxa composing the original metagenome (all Archaeal groups, algal chloroplast sequences, the *Pelagibacteraceae*, and the SAR406 lineages) accompanied by the appearance of a large number of less representative taxa, which shows that the enrichment process was halted at a very early stage, as intended.

## Overall analysis of the functional annotation

Significant shifts between the functional annotation of samples SCD0 and SCDE obtained by EBI Metagenomics were observed.

The analysis revealed somewhat surprising large increases in categories GO:0000103 "sulfate assimilation" and GO:0008272 "sulfate transport." Among the other categories, significantly rising in hit numbers were GO:0016846 "carbon-sulfur lyase activity," GO:

0016705 "oxidoreductase activity, acting on paired donors, with incorporation or reduction of molecular oxygen" and GO:0004497 "monooxygenase activity," GO:0051537 "2 iron, 2 sulfur cluster binding," GO:0006730 "one-carbon metabolic process," GO:0004488 "methylenetetrahydrofolate dehydrogenase (NADP$^+$)" and GO:0008864 "formyltetrahydrofolate deformylase," all of which can be associated to the catabolism of MSA. The latter two may suggest that a selection for organisms that metabolize formaldehyde through the condensation with tetrahydrofolate (rather than with tetrahydromethanopterin) may have occurred. Oddly, though, GO:0046653 "tetrahydrofolate metabolic process" and GO:0046654 "tetrahydrofolate biosynthetic process" were found among the significantly decreased categories. Fittingly, several category associated with oxygenic photosynthesis and GO:0004329 "formate-tetrahydrofolate ligase activity" (incorporation of formate into biomass in anaerobic conditions) also decreased in abundance.

Among the monooxygenase genes significantly increased in number during the enrichment, 5 hits for gene *msuD/ssuD* were found in the assembled metagenomic data of the enriched sample (SCDE), while no homolog was present in sample SCD0.

Since no hits for *msm* genes were observed in the metagenomic results by either EBI Metagenomics or IMG/MER, the sequence data were also screened locally by hmmsearch (HMMER3) and tblastn (*McGinnis & Madden, 2004*). The only hits obtained with protein MsmA (expectation value cutoff $10^{-5}$) contained short Rieske-associated motifs (spacers varying from 16 to 18 in SCD0, and from 16 to 23 in SCDE) and were low scoring. As for protein MsmE, no hits were found at $E$-value $\leq 10^{-5}$.

In order to complement the metagenomic data, *msmA* and *msmE* genes were investigated by an amplicon survey experiment. Direct amplification was weak or absent, so a nested PCR approach was adopted for both genes, which witnesses to the overall low concentration of these sequences in the metagenomes. Amplification of gene A was achieved from both samples. On the contrary, despite repeated and insistent efforts, we were unable to amplify gene E from the enrichment sample (SCDE). For convenience, the three amplicon sets will be designated as follows: SCD0-A (*msmA* amplicons from sample SCD0), SCDE-A (*msmA* amplicons from sample SCDE) and SCD0-E (*msmE* amplicons from sample SCD0). The three amplicon pools were sequenced by ion Torrent technology. The G + C content of the processed reads was close to 40% in all 3 cases (Table S5). Namely, while community metagenome grew from 43.30% to 53.29% G + C content, there was almost no difference in base composition between of the *msmA* reads before and after the enrichment. The reads were then translated and checked for frameshift-generating sequencing errors and all sequences containing an undetermined codon or a stop codon were eliminated. Statistics relative to each case are shown in Table S5.

Rarefaction curves constructed from clustering data at a 0.03 distance are represented in Fig. 4. Although a definite plateau is not achieved, gene A looks sampled at satisfactory depth in both SCD0-A and SCDE-A sets. On the contrary, the level of amplification and sequencing of gene E in set SCD0-E was clearly insufficient. These results suggest that a deeper sequencing would be needed. However, it is also true that a huge part of the data was deleted at the quality control step (Table S5) so better sequencing quality rather than deeper sequencing might be the solution.

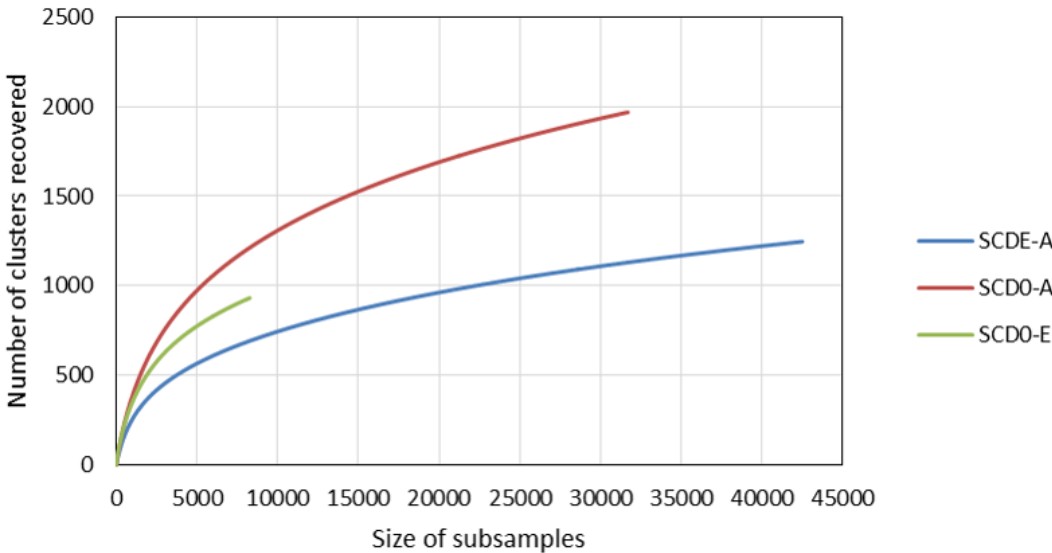

**Figure 4** **Comparison of the rarefaction curves constructed with sequencing data from the amplicon survey experiment.** SCD0-A and SCDE-A refer to gene *msmA* before and after the enrichment with MSA, respectively. SCD0-E refers to gene *msmE* before the enrichment.

**Table 3** **Alpha diversity results for *msmA* and *msmE* genes in samples SCD0 and SCDE.**

| | *msmA* gene | | *msmE* gene |
|---|---|---|---|
| **Indexes** | **SCD0** | **SCDE** | **SCD0** |
| Shannon | 5.08 | 3.56 | 5.32 |
| Evenness | 0.66 | 0.49 | 0.78 |
| $S_{obs}$ | 2,211 | 1,410 | 933 |
| Chao estimator | 3073.2 | 2068.6 | 1331.3 |

The data from clustering of MsmA and MsmE sequences at 0.03 cutoff were analyzed for alpha and beta diversity. Clearly there was an evolution in MsmA sequences during the enrichment process which lead to the loss of some sequence diversity (Table 3). The shift in MsmA allele composition is very significant as illustrated by the high values of the beta diversity indexes (Bray Curtis = 0.95; Jaccard = 0.98; Kulczynski = 0.95; Chao index = 0.76).

Conservation analysis was performed with the predicted MsmA and MsmE sequences (Figs. 5 and 6). The edges of the sequences, corresponding to the PCR primers, should be 100% conserved. The low conservation levels seen in these regions are artifacts due to the occurrence of a few shorter sequences in the datasets and lower quality at the end of reaction. In the case of MsmA (Fig. 5), the levels of amino acid conservation are generally higher and more constant along the central region of the fragment. If the primers regions are not taken into account, all the conservation values are ≥92.35%, with the exception of a one-amino acid trough (a Serine at position 30) with 58% conservation. Particularly, all the amino acids in the Rieske associated motif and spacer demonstrate conservation levels higher than 99%. Regarding MsmE (Fig. 6), it is possible to observe regions with high levels of conservation, which in some cases are greater than 99%. However, the overall

<cHenriques et al. (2016), *PeerJ*, DOI 10.7717/peerj.2498 **12/25**

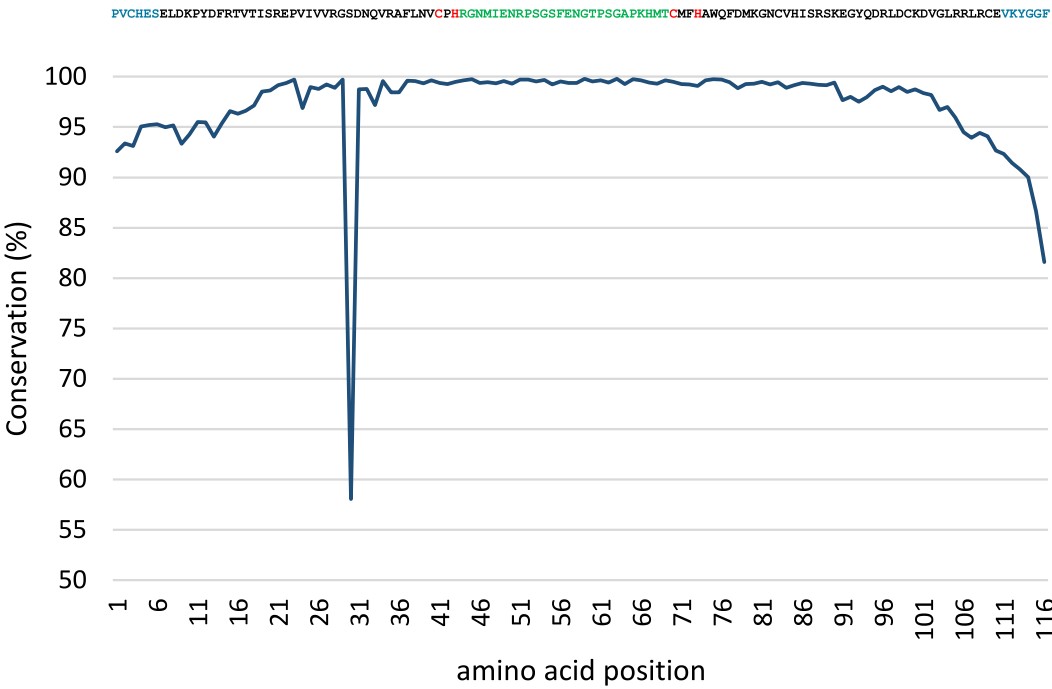

**Figure 5** **Conservation analysis of the predicted MsmA sequences from joined samples SCD0-A and SCDE-A.** Displayed on top is the consensus sequence. Amino acids in blue correspond to PCR primers. Amino acids in red correspond to the cysteine and histidine residues typical of the Rieske-associated motif. Amino acids in green represent the characteristic long spacer found in the Rieske motif in MsmA. Low conservation of the beginning and end of the sequence (corresponding to PCR primers) are artifacts explainable by the presence of short reads in the dataset.

conservation is much less constant than for MsmA, with 4 positions below 85%. The lowest conservation value (61%) was found for a Serine at position 10.

## Phylogenetic trees of the predicted MsmA and MsmE sequences

Phylogenetic trees were constructed based on the nucleotide data corresponding to the representative protein sequences of clusters obtained with distance cutoff values 0.15 (for MsmA) or 0.10 (for MsmE). In the process, sequences from this study were aligned with metagenomic seawater homologs obtained in a previous work (*Henriques & De Marco, 2015a*), with sequences EF103447, EF103448 and EF103448 from SSM clones (*Leitão, Moradas-Ferreira & De Marco, 2009*), with sequences from the GOS project (*Rusch et al., 2007*) and with sequences from cultured strains. Phylograms of *msmA* and *msmE* sequences are shown in Figs. 7 and 8. Although some degree of caution is required in the analysis of these sequences due to the fact that they were produced by three rounds of amplification (whole-genome amplification followed by nested PCR), the results obtained seem to fit logical expectations. In the case of gene *msmA*, it is possible to observe a clear separation (95% bootstrap value) between two principal branches: one consists only of metagenomic sequences and includes all the *msmA* reads generated in this study; the other group (omitted as root in Fig. 7) contains the sequences from cultured strains (*Alpha*, *Beta* or *Gammaproteobacteria*) of both marine and soil origin. In a similar way, the phylogram for

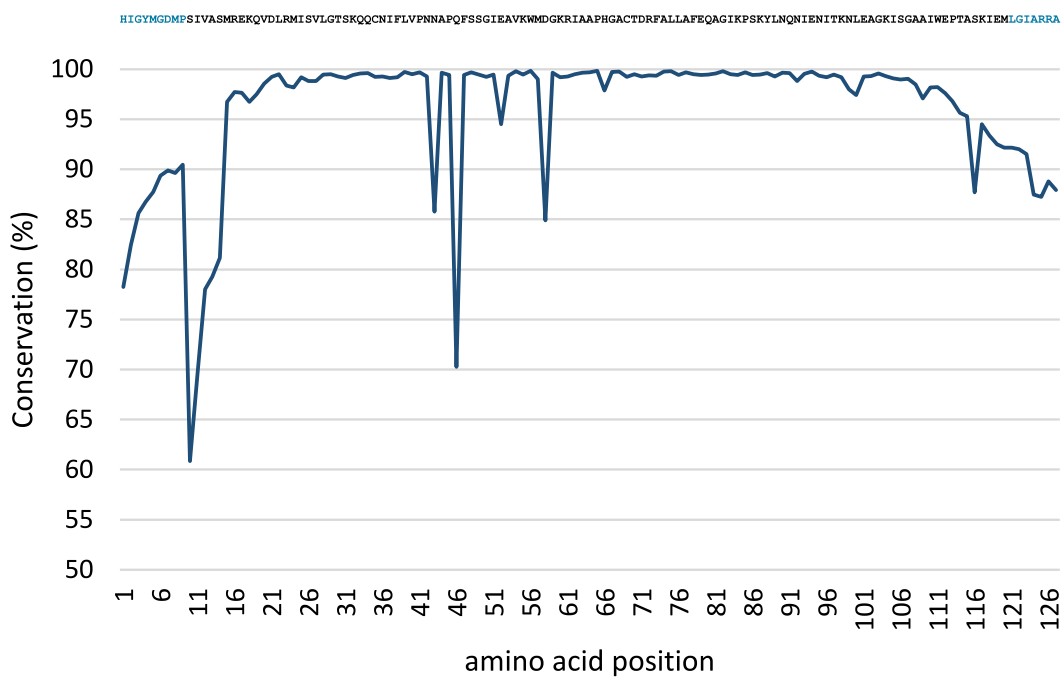

**Figure 6    Conservation analysis of the predicted MsmE sequences from sample SCD0-E.** Displayed on top is the consensus sequence. Amino acids in blue correspond to the PCR primers. Low conservation of the beginning and end of the sequence can be explained as in Fig. 5.

gene *msmE* shows an unambiguous split between two major groups (94% bootstrap value): one comprising mainly metagenomic sequences, including all the *msmE* reads from this study, sequences from the GOS project, and two from cultured marine strains *C. Puniceispirillum marinum* IMCC1322 and *C. Filomicrobium marinum* str. Y; plus a second group containing all other sequences from cultured strains (*Alpha* and *Betaproteobacteria*).

## DISCUSSION

Genes associated to the degradation of methanesulfonic acid have previously been studied in various bacterial isolates and found in several metagenomic studies. A clear G + C-content discrepancy between the former and the latter sequences shows that MSA-degrading strains isolated in the laboratory are not good representatives of the natural communities, especially for marine water. For this reason, in this study we tried to obtain a more faithful snapshot of the MSA-affected microbiota by keeping enrichment parameters as close as possible to field conditions and observing the community at an early stage of enrichment. We started from a surface seawater biomass sample and compared metagenomic information from before and after 16 days of amendment with MSA as the sole organic nutrient. Despite the fact that there was no subculturing, and that the time of enrichment was not sufficient to isolate culturable methylotrophs to purity, the evolution of the community was notorious in the G + C content of the metagenomic data, from 43% of sample SCD0 to 53% of sample SCDE.
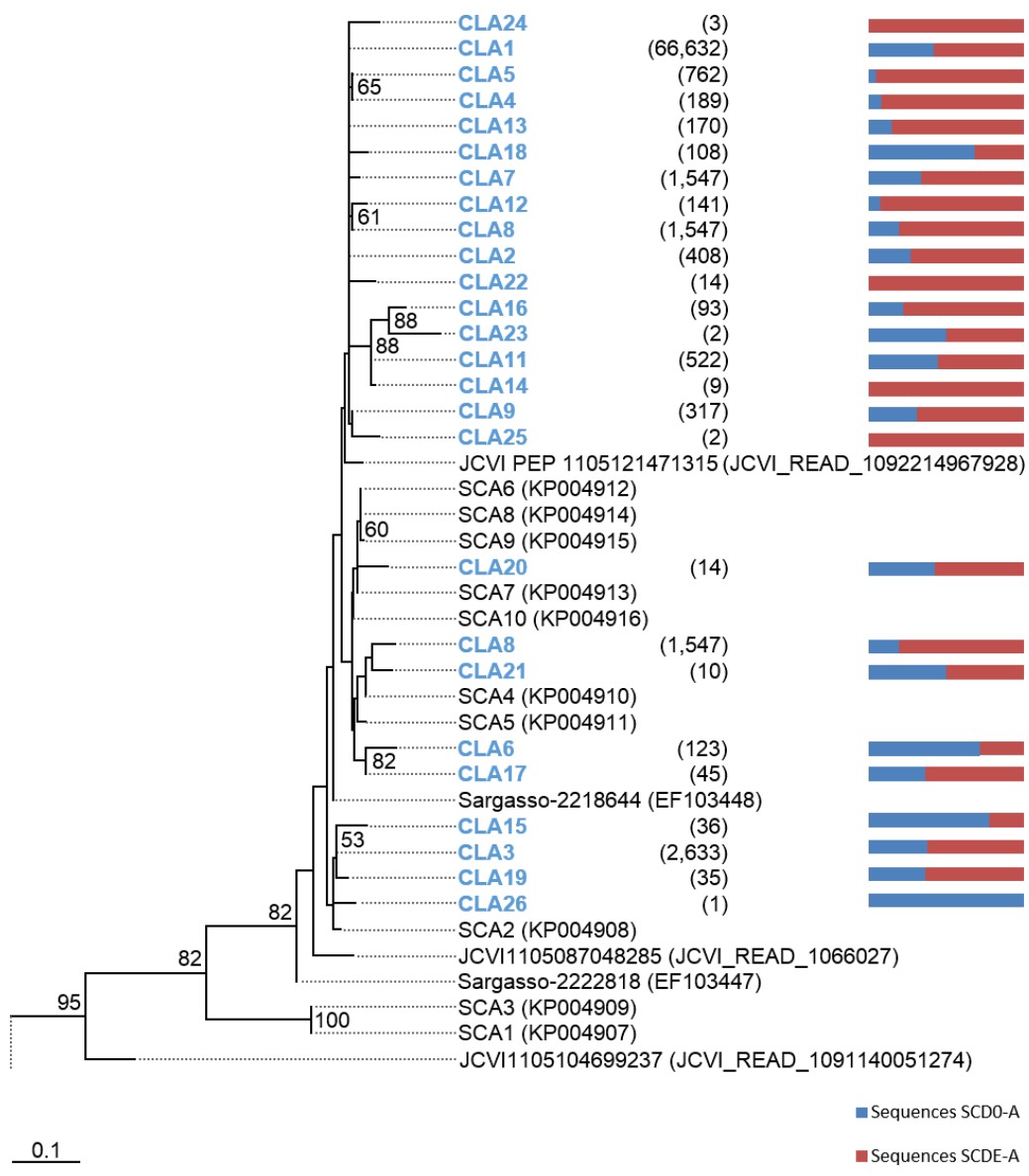

CLA24 (3)
CLA1 (66,632)
65 CLA5 (762)
CLA4 (189)
CLA13 (170)
CLA18 (108)
CLA7 (1,547)
CLA12 (141)
61 CLA8 (1,547)
CLA2 (408)
CLA22 (14)
88 CLA16 (93)
88 CLA23 (2)
CLA11 (522)
CLA14 (9)
CLA9 (317)
CLA25 (2)
JCVI PEP 1105121471315 (JCVI_READ_1092214967928)
SCA6 (KP004912)
SCA8 (KP004914)
60 SCA9 (KP004915)
CLA20 (14)
SCA7 (KP004913)
SCA10 (KP004916)
CLA8 (1,547)
CLA21 (10)
SCA4 (KP004910)
SCA5 (KP004911)
82 CLA6 (123)
CLA17 (45)
Sargasso-2218644 (EF103448)
53 CLA15 (36)
CLA3 (2,633)
CLA19 (35)
82 CLA26 (1)
SCA2 (KP004908)
82 JCVI1105087048285 (JCVI_READ_1066027)
Sargasso-2222818 (EF103447)
95 SCA3 (KP004909)
100 SCA1 (KP004907)
JCVI1105104699237 (JCVI_READ_1091140051274)

■ Sequences SCD0-A

■ Sequences SCDE-A

0.1

**Figure 7  Phylogenetic tree of *msmA* sequences.** Clusters with sequences from this study are in blue: CLA stands for *msmA* clusters. Between brackets is the total number of sequences in each cluster. Horizontal bars indicate the relative frequency of sequences of each cluster (SCD0 in blue and SCDE in red). The accession numbers of the sequences previously published are between brackets. The omitted branch is constituted mostly by sequences from genomes of cultured strains. The accession numbers of these sequences are the following: AF354805 (*Marinosulfonomonas methylotropha* str. TR3), KJ789392 (*Methylobacterium* sp. str. P1), KJ789392 (*Hyphomicrobium* sp. str. P2), GOS sequence JCVI_READ_2101946, KM879220 (*C. Filomicrobium marinum* str. Y), NC_011892 (*Methylobacterium nodulans* ORS 2060), NC_011894.1 (*Methylobacterium nodulans* ORS 2060), KJ789395 (*Methylobacterium* sp. str. RD41), NZ_KB375270 (*Afipia felis* str. ATCC 53690), EF459501 (*Afipia felis* str. 25E1), AF091716 (*Methylosulfonomonas methylovora* str. M2), CP001751 (*C. Puniceispirillum marinum* str. IMCC1322), NZ_AKCV01000022 (*Ralstonia* sp. str. PBA), AP014581 (*Burkholderia* sp. str. RPE67), and CP003775 (*Burkholderia cepacia* str. GG4), CCYE01000041 (*Pseudomonas xanthomarina* str. S11). Nucleotide sequences corresponding to cluster-representative MsmA sequences obtained at 0.15 distance cutoff were used to infer the phylogenetic tree by Maximum Likelihood with 100 bootstrap iterations. Bootstrap values <50% are omitted.

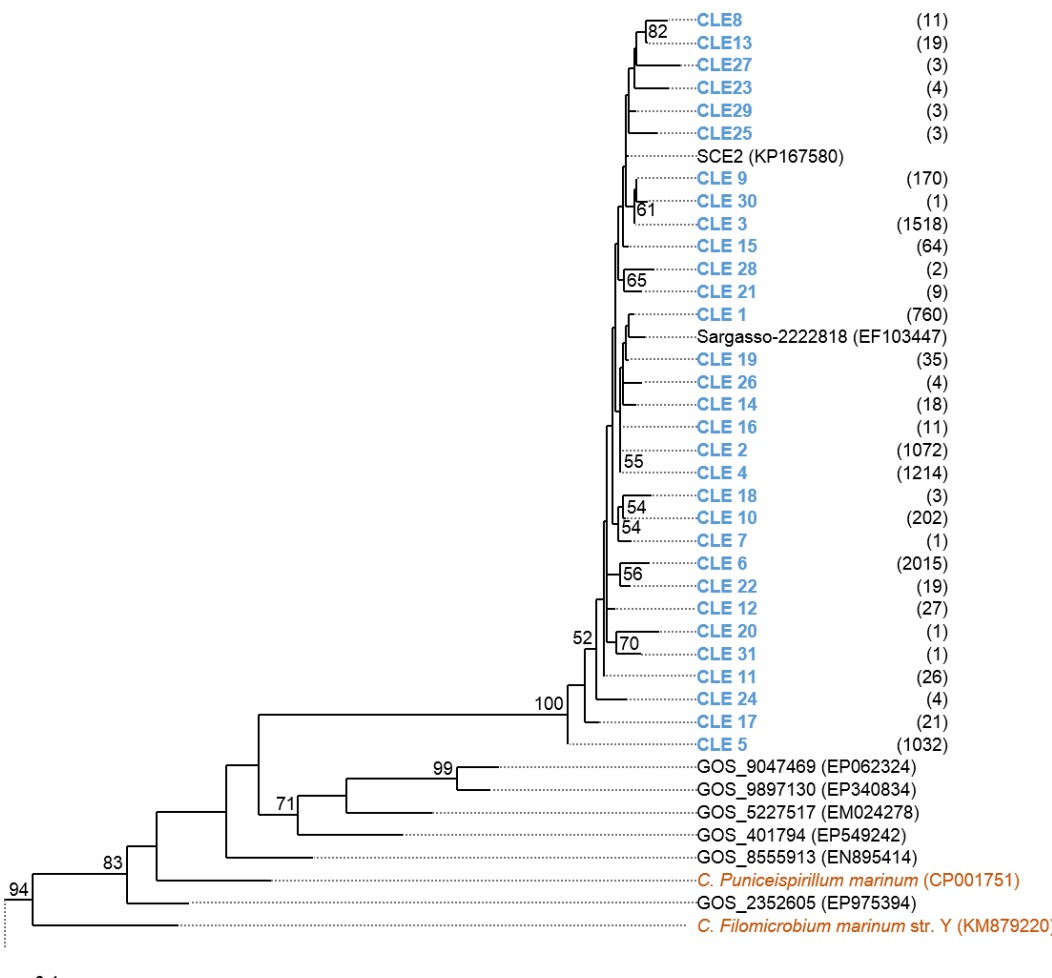

**Figure 8   Phylogenetic tree of *msmE* sequences.** Clusters with sequences from this study are in blue: CLE stands for *msmE* clusters. Between brackets is the total number of sequences in each cluster. Sequences from cultured strains are in orange. The accession numbers of the sequences previously published are between brackets. The omitted branch is constituted by sequences from genomes of cultured strains, with the following accession numbers: NZ_AZUP00000000.1 (*Methyloversatilis discipulorum* str. FAM1), NZ_AFHG01000044 (*Methyloversatilis universalis* str. FAM5), NZ_ARVV01000001 (*Methyloversatilis discipulorum* str. RZ18-153), NZ_AKCV01000024 (*Ralstonia* sp. str. PBA), CCAZ020000001 (*Afipia felis* genospecies A str. 76713), NZ_JNIJ01000008 (*Bradyrhizobium* sp. str. URHD0069), NZ_KB891326 (*Thiobacillus thioparus* str. DSM 505), NZ_AQWL01000003 (*Thiobacillus denitrificans* str. DSM 12475), AZSN01000017 (*Methylibium* sp. str. T29-B), NC_008825 (*Methylibium petroleiphilum* str. PM1), NZ_JADL01000017 (*Rhodospirillales* bacterium str. URHD0088), KP025766 (*Methylobacterium* sp. str. P1), AF091716 (*Methylosulfonomonas methylovora str. M2*), and KP025767 (*Marinosulfonomonas methylotropha* str. TR3). Nucleotide sequences corresponding to cluster-representative MsmE sequences obtained at 0.10 distance cutoff were used to infer the phylogenetic tree by Maximum Likelihood with 100 bootstrap iterations. Bootstrap values <50% are omitted.

Phylogenetically, the enrichment with MSA shifted the prokaryotic community of our sample from a composition fairly typical of oceanic surface waters (mainly *Alphaproteobacteria* (particularly *Pelagibacteraceae*), *Gammaproteobacteria*, *Bacteroidetes*, SAR406 phylum and *Archaea*) to a significantly different community with large increases in methylotroph-harboring taxa such as the *Rhodobacteraceae*, the *Piscirickettsiaceae* and suspected methylotrophs such as *Alcanivorax*, accompanied by other emerging taxa not known to harbor methylotrophs (*Alteromonadales*, *Oceanospirillaceae*, *Hyphomonadaceae*). Such a considerable change in phylogenetic profile can be the result of adaptation to using MSA as nutrient, but may also be the sign of a move from an oligotrophic to a more copiotrophic community.

The indications obtained from the analysis of the metagenomic sequences at the functional level showed enrichment in some functions associated with methylotrophic metabolism and in [2Fe-2S] cluster-containing monooxygenases, a category including MSAMO.

Binning of the metagenomic sequence data revealed the presence of three coherent sets of sequences in sample SCDE matching genera whose abundance was seen increasing in the 16S-based phylogenetic analysis: *Alcanivorax*, a possible methylotroph, *Hyphomonas* and *Methylophaga*, a genus of genuine methylotrophic strains. Genus *Hyphomonas* is not known to be able of methylotrophic growth. However, genes possibly involved in $C_1$ metabolism were found in the three bins.

Interesting data were also obtained on the two genes associated to the catabolism of MSA, *msmA* and *msmE*, by a targeted single-gene high-throughput sequencing strategy. Gene *msmA* could not be detected directly in the metagenomic sequence data probably due to an unsuitable ratio between sequencing depth and community complexity. However, after PCR amplification and sequencing, a multitude of diverse sequences was obtained from both the original and enriched sample: despite the shifts in phylogenetic composition caused by the enrichment process, at the level of gene *msmA* the G + C content remained practically constant. All these sequences from both samples were shown to belong in a very solid metagenomic branch, clearly separate from all the homologs encountered in cultivated strains of both soil and marine origin and unmistakably some sequence clusters were seen shrinking (or even disappearing) or expanding due to the enrichment regime. Gene E too was not found in the metagenomic sequence data of either sample; however, we could amplify it from the initial seawater sample (SCD0) with a variety of diverse sequences most closely related to other known metagenomic sequences. Also in this case, our *msmE* sequences together with other metagenomic occurrences formed a group clearly distinct from homologs encountered in soil species, although in this instance clustering with the *msmE* genes from two marine cultivated strains (C. *Puniceispirillum marinum* and C. *Filomicrobium* str. Y) was observed. Rather surprisingly, we were not able to amplify gene E from the enriched sample (SCDE). This fact suggests that two different populations of bacteria may have existed in sample SCD0: one, harboring both genes A and E, which was wiped out during the enrichment, and another carrying just the MSA monooxygenase (gene A) and importing methanesulfonate from the medium using a transporter alternative to MsmE whose gene cannot be amplified by the primers we employed. Scanning through

the functional annotation of the two assembled metagenome sequences, we found 31 "ABC-type nitrate/sulfonate/bicarbonate transport system, periplasmic component" hits in SCD0 and 55 in SCDE: this is the description of a family of MsmE paralogs. It was already recognized that gene E showed a lesser level of conservation than gene A and our data corroborate this idea and indicate that regrettably *msmE* cannot be used as a reliable functional indicator of MSA utilizers.

An alternative enzyme has been found in non-methylotrophs that use MSA as a source of sulfur: SsuD (also MsuD) is an $FMNH_2$-dependent alkanesulfonate monooxygenase which can desulfonate MSA (*Eichhorn, Van Der Ploeg & Leisinger, 1999*; *Endoh et al., 2003*). Intriguingly, we recently described a *Rhododoccus* strain that grows on MSA possibly employing SsuD (*Henriques & De Marco, 2015c*). The fact that we detected 5 *msuD*/*ssuD* homologs in the enriched metagenome (and none in the original sample) suggests that this alternative sulfonate monooxygenase may be another relevant marker gene for MSA utilization in the oceanic environment.

This evidence may also point to two possible alternative scenarios: in the first, at least part of the degradation of methanesulfonate is accomplished by non-methylotrophic bacteria capable of cleaving the MSA molecule and of using the resulting formaldehyde purely as a source of energy: a similar model has been found at work with $C_1$ compounds such as methanol, formaldehyde, methylamine, trimethylamine, trimethylamine N-oxide and the $C_1$ moiety of glycine betaine and dimethylsulfoniopropionate in marine strain *C. Pelagibacter ubique* HTCC1062 of the SAR11 clade (*Sun et al., 2011*) which lead the authors to coin the term 'methylovore' for this type of metabolism. In that study, MSA was not a tested compound: indeed, genes *msmA* or *ssuD* are not present in the published genome sequences within the *Pelagibacteraceae* and we see this family disappear from our enriched sample. However, it is not implausible that other non-methylotrophic marine species may use MSA co-metabolically as a source of energy in an analogous fashion. A second scenario can be imagined of a syntrophic association between non-methylotrophic microorganisms expressing MsmA or SsuD and methylotrophs incapable of cleaving MSA. This hypothesis is especially relevant for marine water where many of the microorganisms are found in suspended particles or flocks (*Azam, 1998*).

Due to time and practical constraints, this study reports on the community evolution of a single specific sample. As such, the influence of confounding factors on the outcomes cannot be properly discarded and our results must be considered explorative in nature. However, many of the phylogenetic and metabolic data obtained distinctly suggest that MSA was the factor genuinely causing the differences observed. Gene *msmA* appears to be a good marker for MSA degraders, although further study into organisms employing alternative MSA-cleaving enzymes is needed in order to assess the relative importance of the different genes. Enrichment-free strategies such as single-cell genome sequencing, which provides joined functional and phylogenetic data, or MSA-induced metatranscriptomic shift analysis or *in situ* physiological studies are necessary to obtain a more complete and realistic picture less dependent on lab culturing.

**Abbreviations**

| | |
|---|---|
| **MSA** | Methanesulfonic acid |
| **MSAMO** | MSA monooxygenase |
| **DMS** | Dimethylsulfide |
| **SSM** | Sargasso Sea metagenome |
| **GOS** | Global Ocean Sampling |

## ACKNOWLEDGEMENTS

We wish to acknowledge the help received from colleagues at CESPU Patrícia Duarte, Nilza Ribeiro, Claudia Ribeiro and Hugo Ribeiro, and Hassan Bousbaa (IINFACTS) for institutional support. We also thank Ana Toribio (ENA), Hubert Denise (EBI Metagenomics), Benli Chai (RDP), Jennifer Jackson (Galaxy), Matthieu Vizuete-Forster (Illumina), Scot Dowd (Molecular Research LP), the NGS team at Stabvida, I-Min Chen (IMG/JGI), Brandon KB Seah (Max Planck Institute for Marine Microbiology, Bremen, Germany), and Inês Cruz (Faculty of Science, U. of Porto, Portugal) for their patient technical advice. Finally, we wish to acknowledge the helpful criticism by the two anonymous reviewers of this work.

### Funding

This work was funded by the European Commission's Regional Development Fund (ERDF) through the Competitiveness Factors Operational Program (COMPETE - project FCOMP-01-0124-FEDER-028330) and by the Portuguese Science Fund (FCT - project PTDC/BIA-MIC/3623/2012). The funders had no role in study design, data collection and analysis, decision to publish, or preparation of the manuscript.

### Grant Disclosures

The following grant information was disclosed by the authors:
European Commission's Regional Development Fund (ERDF): FCOMP-01-0124-FEDER-028330.
Portuguese Science Fund: PTDC/BIA-MIC/3623/2012.

### Competing Interests

The authors declare there are no competing interests.

### Author Contributions

- Ana C. Henriques conceived and designed the experiments, performed the experiments, analyzed the data, contributed reagents/materials/analysis tools, wrote the paper, prepared figures and/or tables, reviewed drafts of the paper.
- Rui M.S. Azevedo conceived and designed the experiments, performed the experiments, analyzed the data, contributed reagents/materials/analysis tools, wrote the paper, reviewed drafts of the paper.

- Paolo De Marco conceived and designed the experiments, performed the experiments, analyzed the data, wrote the paper, prepared figures and/or tables, reviewed drafts of the paper.

## DNA Deposition

The following information was supplied regarding the deposition of DNA sequences:

DOE Joint Genome Institute's Integrated Microbial Genome Metagenomic Expert Review (IMG/MER) GOLD Project ID Gp0111927: (biosamples Gp0111627 and Gp0111630).

European Nucleotide Archive project number PRJEB9018 (sample accession numbers ERS700852, ERS700853, ERS954926, ERS954925 and ERS954927).

## Supplemental Information

Supplemental information for this article can be found online at http://dx.doi.org/10.7717/peerj.2498#supplemental-information.

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
