# Peer review of "Metagenomic survey of methanesulfonic acid (MSA) catabolic genes in an Atlantic Ocean surface water sample and in a partial enrichment"

_PeerJ, doi:10.7717/peerj.2498_

## Round 0.1 · original submission · Major Revisions

Both reviewers found merit in your study, and were very supportive of research into microbial populations involved in MSA degradation in marine ecosystems. However, they also both identified several issues related to data analysis & presentation, as well as experimental set-up with respect to the obvious lack of biological replicates.

Reviewer 1 ·

Basic reporting

no comment

Experimental design

no comment

Validity of the findings

fine

Additional comments

The manuscript submitted by Henriques et al. presents an interesting study on the diversity of bacteria in the Atlantic Ocean with the potential to use methanesulfonic acid (MSA) as a growth substrate. The study is relatively well conceived. They use an enrichment strategy, amending seawater with MSA and tracking shifts in community composition and genomic content using PCR-based and metagenomic approaches. The researchers conclude that there are numerous methylotrophic bacteria in the ocean that possess the MSA monooxygenase genes required for MSA oxidation, but that the msaE transport system is not as common, suggesting alternative MSA transporters in the marine community.
The introduction is well written, the methodology is appropriate, and the metagenomic data is analyzed properly using some of the latest methods.
However, the results and discussion section could really use some work. Here are some suggestions:

1. Most of lines 233-246 are either redundant with the methods, or could just be put in the methods
2. There seem to be some errors in Figure 3. For example, there are two entries for strain HTCC2188. Why is that? Also there’s no label on the x axis.
3. Line 302. Why was the enrichment process halted at a very early stage and why was this intended? Also is 16 days actually a short incubation? Seems quite long to me? A clarification on what is meant be early stage would be helpful
4. The metagenomes were assembled, so it would be nice if the researchers did some binning of contigs to particular taxonomic groups using coverage distribution across the two samples as well as tetranucleotide frequencies (lots of software that will do this), and the did some metabolic reconstruction on the most variable populations impacted by enrichment. This may provide very novel information on the MSA effects on the microbial community.
5. The researchers did whole genome amplification followed by nested PCR to detect MSA! So it’s really at low copy number I guess. They should acknowledge this in their discussion
6. How are the Beta diversity estimates in Table 4 and elsewhere informative? Seems like they could be removed to me.
7. What is the point trying to be made with Figure 5 and 6? They seem rather useless too, and the phylogenetic trees make the point about relationships
8. A reference at line 458 is required
9. Line 462, was MSA really kept at close to field conditions? 5 mM is a natural concentration?
10. Line 469-497. There’s mix ups here and throughout the text on the taxonomic groups. Here for example: Pelagibacter are Alphaproteobacteria, so why write Alphaproteobacteria AND Pelagibacter
11. Is there a possibility of crossfeeding occurring in the incubations? Could be interesting to discuss.

Reviewer 2 ·

Basic reporting

The manuscript by Henriques et al. provides genetic and metagenomic evidences for microbial communities involved/contribution to methanesulfonic acid (MSA) cycling in sea water. Overall the manuscript well written and easy to follow. Sufficient background information was provided.

Experimental design

The experimental design is well described. If available, the addition information on in-situ concentrations of MSA, rates of MSA consumption or at least final concentrations of MSA in enrichment cultures should be provided.
The manuscript lacks information for control experiments for WGA studies.

Validity of the findings

That is a descriptive study, which characterized microbial communities enriched on MSA. Main concern- the use of one biological replicate for microcosm experiments.

Additional comments

MSA is potentially one of the largest sources of carbon in marine ecosystem. While some metabolic pathways for microbial MSA utilization are established (e.g. msm- enzymes), the overall knowledge of microbial communities involved in the organic-S compound cycling in natural environments is highly fragmented. The authors took a short-term enrichment (microcosms) approach to investigate the structure of microbial community responding to MSA. Overall, the study provides additional insights into the diversity of microbes contributing to MSA degradation in oceanic surface waters.
General comments:
1. The concentration of MSA used in the spike experiments seems to be high (total amount added is 20mM). Are there any reason to use very high carbon input for typically nutrients limited system? Were toxicity issues considered? Please provide data that would indicate that the used MSA amounts are not toxic to other members of the community (no spontaneous lysis of cell due to toxicity over the incubation period).
2. According to method description only one biological replicate was set-up. If it is correct, the authors should justify why they used only one biological sample.
3. msm- diversity after at least three rounds of amplification (Repli-G/ first nested PCR and second PCR), are those valid? This is a minor part of the manuscript and I would recommend to remove this part.

Minor comments:
L103. Add conductivity data to results (L231-232).
L109. What was the initial cell density? And how it changed over the incubation period?
L157. Remove extra space after …. 2015).
L231. Should it be 3.34%?
L232. What were the final concentrations of chlorophyll?
L248. Move Figure 1 to supplemental materials.
L283. Figure 2 and 3. Define “Root” or remove. Consider to cluster groups by as Alpha or Gamma or Delta?

---

## Round 0.2 · Minor Revisions

The revised manuscript has certainly much improved as compared to the original paper, however, the main issue related to the lack of replication remains. Lack of replication in principle means that you can never be sure what the reason is for the observed differences. is it the incubation, the fact that samples are from two separate filters, is it because of.....

This should be made more explicit in a revised manuscript.
Lack of funds is always a serious issue most of us face when planning experiments. In this case, for example, it might have been a better idea to at least have replicate enrichment cultures starting from one or multiple samples, and then do the amplicon sequencing of specific genes on multiple samples, in addition to the single sample full metagenome survey. Now there is basically no replication at all for any of your data.

In addition to the reviewers' suggestions, I would like to add some from my side.

1) Please avoid the terms flora or microflora. Microbes are not plants, and should be referred to as e.g. microbiota.

2) Regarding taxonomic assignment of sequences, it is indeed an issue that different sequences/OTUs are classified at different levels of taxonomic resolution (e.g. Pelagibacter vs. unclassified Alphaproteobacteria). That should be made more explicit in the text by referring e.g. to Pelagibacter and other, unclassified members of the Alphaproteobacteria, rather than "Pelagibacteraceae and Alphaproteobacteria" as is done now.

Reviewer 1 ·

Basic reporting

No comments

Experimental design

No comments

Validity of the findings

No comments

Additional comments

Numbering of comments is as flows the author’s rebuttal letter

2. This is a nonsensical explanation: What is meant by “family HTCC2181” and “genus HTCC”? it’s not good enough to say this is the output of an automatic classification. The author’s need to manually curate the output if they strive for accuracy. Alos, there is still no X axis title given for this figure.

3. 16 days and 2 months doesn’t seem to be that different to me. Short to me would mean 24 hours or less, microbes could absolutely respond over such a “short time frame”. I still recommend the author’s remove any mention of “short” in the manuscript

7. I agree with the author’s but you still don’t need these figures to make the point. I’ll leave it up to the editor to decide.

Reviewer 3 ·

Basic reporting

The manuscript describes initial investigation of the methanesulfonic acid (MSA) degradation in oceanic water sample via metagenomic and enrichment studies. While the study uncovers some possibly novel mechanisms for MSA catabolism and suggests novel players in MSA cycling, the overall experimental design of the study has some flaws.

Experimental design

The authors addressed all my minor comments. However, the main concern, "a single replicate study" design was not (and could not be) addressed.

Validity of the findings

(see main concern above).

Additional comments

All results presented in the study are still based on only one replicate. In my opinion any one-replicate metagenomic study could be used only as preliminary data. However, as various journals follow different rules now, I would like to let the editor to make a decision in accordance with the journal policies.

---

## Round 0.3 · accepted · Accept

You have responded appropriately to all issues raised by the reviewer, and also indicated now that the study described in the paper is of exploratory nature due to the limited sample size.